# Genome-Wide Identification of ARF Gene Family Suggests a Functional Expression Pattern during Fruitlet Abscission in *Prunus avium* L.

**DOI:** 10.3390/ijms222111968

**Published:** 2021-11-04

**Authors:** Qiandong Hou, Zhilang Qiu, Zhuang Wen, Huimin Zhang, Zhengchun Li, Yi Hong, Guang Qiao, Xiaopeng Wen

**Affiliations:** 1Key Laboratory of Plant Resource Conservation and Germplasm Innovation in Mountainous Region (Ministry of Education), Collaborative Innovation Center for Mountain Ecology & Agro-Bioengineering (CICMEAB), Institute of Agro-Bioengineering/College of Life Sciences, Guizhou University, Guiyang 550025, China; qiandhou@163.com (Q.H.); 18786621377@163.com (Z.Q.); gzu_zwen@163.com (Z.W.); hongyi715@163.com (Y.H.); 13518504594@163.com (G.Q.); 2College of Forestry, Guizhou University/Institute for Forest Resources & Environment of Guizhou, Guiyang 550025, China; mhz722@126.com (H.Z.); zcliah@163.com (Z.L.)

**Keywords:** genome-wide, *Prunus avium*, auxin response, gene family, fruitlet abscission

## Abstract

Auxin response factors (ARFs) play a vital role in plant growth and development. In the current study, 16 ARF members have been identified in the sweet cherry (*Prunus avium* L.) genome. These genes are all located in the nucleus. Sequence analysis showed that genes in the same subgroup have similar exon-intron structures. A phylogenetic tree has been divided into five groups. The promoter sequence includes six kinds of plant hormone-related elements, as well as abiotic stress response elements such as low temperature or drought. The expression patterns of *PavARF* in different tissues, fruitlet abscission, cold and drought treatment were comprehensively analyzed. *PavARF10/13* was up-regulated and *PavARF4/7/11/12/15* was down-regulated in fruitlet abscising. These genes may be involved in the regulation of fruit drop in sweet cherry fruits. This study comprehensively analyzed the bioinformatics and expression pattern of *PavARF*, which can lay the foundation for further understanding the *PavARF* family in plant growth development and fruit abscission.

## 1. Introduction

Auxins are a kind of phytohormone widely found in plants, which are influential for plant growth and development phases, including seed germination, inflorescence, fruit development, leaflet formation and blade growth, root architecture and differentiation, etc. [1]. Through binding to receptor cell membranes or intracellular components and interacting with other signal transduction pathways, auxins can directly regulate the processes of cell division, differentiation and elongation, and regulate development [2]. Auxin signal transduction entails transcriptional activation of auxin-regulated genes and degradation of transcription repressor protein, this process show variability and complexity [3]. There are many gene families involved in auxin signal transduction, including auxin response factors (ARFs), auxin/indole-3-acetic acid (AUX/IAA), small auxin-up RNAs (SAURs), Gretchen Hagen3 (GH3), etc [3]. In plants, auxin levels are not immutable, and changes in auxin level frequently trigger transient changes in these gene families to regulate plant growth and development as well as the impact of variations in the external environment [4]. There are two types of transcription factor families that play a crucial role in the expression of auxin-responsive genes. One is ARF, which can activate or inhibit the expression of the target gene by combining with the auxin response element (AuxRE) with a special sequence TGTCTC on the promoter of the target gene; the other is AUX/IAA, is homologous to ARF, functions by binding to ARF and degradation of it [5,6].

One integral ARF generally consists of three conservative domains. The amino-terminal has a B3-type DNA-binding domain (DBD), and the middle region (MR) is partitioned according to the amino acid type into an activation domain or an inhibition domain and also has a carboxy-terminal dimerization (CTD) domain [7]. Crystal structures of the DBD demonstrated three different subunits, one which allows ARF to be assembled a biologically active dimerization domain; a B3 type domain that was highly similar to the DNA binding domain in prokaryotic endonucleases and was plant-specific; and a Tudor-like ancillary domain whose function is not yet clear [8]. DBD binds to AuxRE by the B3 on the auxin-responsive gene promoter that contains TGTCTC motif. According to the specific amino acids contained in MR, ARF proteins classified into three subgroups, class A was considered to be a transcriptional activator and was rich in glutamine (Q) and a small amount of leucine (L) and serine (S) residues; the remaining ARF is rich in serine, threonine (T) and proline (P) as transcriptional repressors, and perchance branched into the *miR160*–targeted ARFs (class C) and the rest of ARFs (class B) [9]. The carboxy-terminal domain does not include all ARFs and plays an important role in mediating the interactions between ARF and AUX/IAA [10].

By and large, when auxin concentrations are low, AUX/IAA will combine directly with ARFs. AUX/IAA will recruit a TOPLESS (TPL) or TOPLESS related (TRR) co-repressor to promote the deacetylation of histones, making the chromosomes compressed to reduce the accessibility of DNA, and achieve the purpose of inhibiting gene expression [4]. As the concentration of auxin increases, AUX/IAA binds to the TRANSPORT INHIBITOR RESISTANT1/AUXIN SIGNALING F-BOX (TIR1/AFB) protein family by the induction of auxin. These TIR1/AFB proteins have one F-box E3 ubiquitin type ligase that is recognized by SCF-type ubiquitin protein ligase complexes, which triggers the polyubiquitination and degradation of AUX/IAA, and ARFs are released. ARFs will recruit SWITCH/SUCROSE NONFERMENTING (SWI/SNF) to change the chromatin state, and then combine with the AuxRE element on the promoter of the target gene to start the expression of the auxin-responsive [10]. Thus far, in the *Arabidopsis* genome, 23 *AtARF* members have been listed [11]. All of these AtARF3/13/17/23 proteins are incomplete and lack the CTD domain [5]. In response to some internal and external environmental changes, the above variations in diverse genetic domains may have distinct functions in the auxin signaling pathway and play a non-negligible role. After more than two decades of research, the signaling pathway induced by auxin mechanism has been largely elucidated.

The ARF gene family plays an important role in plant resistance. *AtARF7* have different expression patterns in dry and humid environments, which determines the plant’s construction of a root branching model in response to water availability [12]. In tomato, *SlARF1*, *SlARF4*, *SlARF6B*, *SlARF10A* and *SlARF18* were significantly up-regulated in response to drought stress, while *SlARF3*, *SlARF5*, *SlARF6A*, *SlARF7A* and *SlARF19* ware significantly decreased [13]. When mulberry was treated in desiccation conditions, 13 ARFs have shown differential expression, suggesting that ARF can be widely involved in regulating drought stress [14]. The expressions of nine *SbARF* genes (*SbARF4/7/9/15/17/ 19/21/22/24*) increased after cold treatment in *Sorghum bicolor*, while *SbARF13* remained decreased during the whole treatment [15]. Most of the 25 *OsARF* members in rice can be induced by drought or low temperature, and *OsARF4/11/13/14/16/18* can be induced in at least one condition [16].

So far, the ARF family have been identified from many species such as *Oryza sativa*, *Zea mays*, *Prunus persica*, *Brassica napus*, soybean and so on and their functions have also been researched in depth [17,18,19,20,21]. ARFs regulate a diverse range of plant behaviours as transcription factors, including development processes and abiotic stress response. In wheat, *TaARF8*, *TaARF9*, and *TaARF2* expression were significantly altered by low temperature in genic male-sterile, and these genes were involved in the cold-induced male sterility pathway [22]. *IbARF5* in sweet potato participates in the synthesis of carotenoids and plays an important regulatory role in salt and drought stress [23]. The expression level of *LcARF2D/2E*, *7A/7B*, *9A/9B*, *16A/16B* in litchi changes significantly during fruit abscissing, and *LcARF5A/B* has a positive regulatory effect during fruit abscising [24].

Sweet cherry (*Prunus avium* L.) is extensively cultivated as an economic and horticultural fruit tree in southwestern China. Throughout flower development, they are susceptible to spring frost and prone to abnormal fruit abscission during fruit development. The sweet cherry *PavARF* gene family members have not been reported previously, and their functions remain to be elucidated under cold and drought stress conditions. The completion of the whole genome sequencing of sweet cherries makes it possible to study ARF gene family [25]. Genome-wide identification of gene families and analysis of their functions have been studied in other species. For example, the abiotic stress function of LncRNAs in *Capsicum annuum* [26], the AHL gene family in soybean [27], and the expression characteristics of KCS gene family in barley development [28], and so on. Therefore, this study identified and analyzed ARF gene family by searching the whole genome of sweet cherry and using bioinformatics methods and included tissue-specific expression profile and dynamic expression patterns in response to cold and drought stress, which can provide a meaningful reference for further functional investigations of the *PavARF* gene family in fruit abscission and abiotic stress.

## 2. Results

### 2.1. Identification of ARF in Sweet Cherry

An HMM profile of the ARF was employed as a query to identify the ARF genes in sweet cherry. Finally, a total of 16 *PavARF* genes were identified in the sweet cherry genome. Following the nomenclature for members ARF, these genes were successively named based on chromosomal position. The remaining genes that cannot be located on the chromosome and located on random scaffolds of the ‘chr 0′ reference sequence are named sequentially according to their gene ID. These members of the *PavARF* gene family are summarized in Table 1. The length of PavARFs aa (amino acids) ranged from 384 (PavARF3) to 1187 aa (PavARF1), and the molecular weight ranged from 42.8 to 119.6 KDa. Besides, the isoelectric points (pI) of the 16 ARF proteins were also predicted to range from 5.20 to 6.44. The GRAVY showed that all ARF are hydrophilic proteins. The prediction of subcellular localization showed that 16 PavARF were localized in the nucleus. The 11 *PavARF* members have complete domains. Interestingly, two ARF genes related to fruit development were found in the’ Tieton’ genome, among which FUN_011630-T1 and *PavARF4* sequence are identical, while FUN_032112-T1 and *PavARF7* sequence are highly similar [29].

### 2.2. Phylogenetic Analysis of PavARF

To know more about the conserved domains in PavARF, the PavARFs were aligned (Appendix A). The results show that these sequences have a certain degree of similarity, and two highly conserved domains (DBD and ARF Domain) were identified. However, not all the PavARFs contain a CTD domain, such as *PavARF5/7/12/13*, while the DBD domain was not found in *PavARF3*. Phylogenetic analysis involving the deduced protein sequences of the 16 PavARF, 17 PpARF of the *Prunus persica* (peach) [17] and 31 MdARF of *Malus domestica* (apple) [30] was performed. The phylogenetic tree of ARF proteins was divided into four classes (clade A1, clade A2, clade A3 and clade A4) (Figure 1). Most of the members of *PavARF* were distributed in the cladeA1 subgroup, and the four members that activate transcriptional activation belong to the cladeA1 subgroup. Among the 16 PavARF proteins, 5 belong to clade A1, 4 to clade A2, 2 to clade A3, and clade A4 have 5. Genes in the same subfamily may have evolved from the same gene duplication event, and their functions may also be similar.

### 2.3. Gene Structure and Motif Analysis

In order to understand the gene structure and conserved motifs of ARF, the intron-exon structure of these members was obtained through the genomic DNA sequence. The phylogenetic tree was constructed with the NJ method and 1000 bootstraps for the full length of the 16-member protein in MEGA7.0. The results showed that 16 motifs were found in these members, but there were differences in the number of motifs between genes. (Figure 2b). There are only five motifs to the *PavARF3*, which may be the reason for the lack of the B3 domain. None of the four genes lacking the CTD domain have motif 14. Motif 16 was a special conserved motif, which only existed in *PavARF6/8/16*. The interesting creature was that they are clustered into one category in an evolutionary relationship. *PavARF3* lacks the *N*-terminal B3 domain, and the DBD domains of the other 15 members are highly conserved. The Auxin_resp domains of all members are quite different, which is also the area where ARF functions. The number of exons differed from three to 14 in *PavARF* genes, thus, the general gene structure of this family is complex and diverse (Figure 2c). Like ARF of other species, *PavARF* can be divided into three groups: *PavARF1/6/8/14/16* consituted group A, *PavARF2/5/12/13* belong to group C, A and C were clustered in CladeA1; group B contained CladeA2-5, including *PavARF3/4/7/9/10/11/15.* Furthermore, the same clade for *PavARF* members usually displayed similar exon-intron arrangements. The middle region of *PavARF* is counted (Figure 2d). *PavARF1/6/8/16* has potential transcriptional activation activity, and their middle region was Q-rich; members with transcriptional repressive activity were *PavARF4/9/10/11/12/13/15*, in the middle was PST-rich. Based on the types of amino acids in the middle region of ARF, their characteristics can be predicted to a certain extent, but the real situation needs to be further verified by experiments such as transient expression of protoplasts. *PavARF12/13* has lacked the CTD domain.

### 2.4. Analysis of Hormone-Related Cis-Elements

The promoter region upstream of the gene is very vital during gene transcription, and generally has distinct functions, including abiotic stress and response to phytohormone. To better understand the *cis*-acting elements of *PavARF* gene promoters, the upstream sequences of all identified *PavARF* were submitted and calculated by Plant-CARE. The prediction results showed that in the upstream region of *PavARF*, in addition to some core promoter regions such as TATA-box and CAAT, it also includes elements such as light response and abiotic stress (Appendix A).

Because auxin response factors are mainly related to phytohormone, mainly select elements related to phytohormone and respond to low temperature for statistics (Figure 3). Interestingly, the elements that respond to drought are concentrated within 1000 bp downstream, and this distribution may be more conducive to gene response to drought stress.

According to predictions, a total of six hormonal response-related elements were found, which were auxin responsive (TGA-element, AACGAC and AuxRR-core, GGTCCAT, gibberellin responsive (CCTTTTG), abscisic acid responsive (ACGTG), MeJA (methyl jasmonate, CGTCA) responsive, ethylene responsive (ATTTTAAA) and salicylic acid responsive (TCAGAAGAGG/CCATCTTTTT). Additionally, two stress-responsive regulatory elements, CCGAAA-motif associated with low-temperature responses (LTR) and CAACTG associated with drought-inducibility was identified in the *PavARF* promoter regions. These elements were unevenly distributed upstream of 16 *PavARF* genes. *PavARF11* had only one ethylene response element; *PavARF2* had 17 response elements, including five different response elements. *PavARF9* contained the most ACGTG motifs related to abscisic acid response, suggesting that might be played a role in the abscisic acid signaling pathway. The selected seven response elements were included in *PavARF4*, implying it could engage throughout the response to different stress and hormone treatments via the various regulatory mechanisms. 

### 2.5. Genomic Distribution and Gene Duplication

Based on the genes coordinate annotation data, we mapped identified sweet cherry ARF on chromosomal. Most genes can be located on chromosomes and their distribution is uneven (Appendix A). Due to the so-called “0” chromosome in the sweet cherry genome [25], the level of genome assembly is limited, and *PavARF16* cannot be located on the chromosome. Chromosome 1 and chromosome 6, which were the two chromosomes with the most *PavARF* genes, each contained three members, while chromosomes 2, 4 and 5 each contained two genes, and chromosomes 3, 7 and 8 had the lowest number of members (only one *PavARF* gene). *PavARF1/2/3/4/5/9/10/13/14* was located on the positive strand of the chromosome, while *PavARF6/7/8/11/12/15* was located on the reverse strand of the chromosome.

To further explore the phylogenetic relationship between sweet cherry and other plants, we analyzed the synteny relationships between six plants and sweet cherry (Figure 4). The six plants include the dicotyledonous plant *Arabidopsis thaliana* TAIR10, soybean (*Glycine max*, v2.1), tomato (*Solanum lycopersicum*, SL3.0), sweet orange (*Citrus sinensis*, v2.0), apple (*Malus domestica*, ASM211411v1.48) and peach (*Prunus persica*, NCBIv2), and the monocotyledonous plant rice (*Oryza sativa,* IRGSP-1.0.47). In these six species (rice, *Arabidopsis*, soybean, tomato, sweet orange, apple and peach), the number of gene pairs with orthologous pairing is 8, 13, 44, 23, 17, 34 and 20, respectively. The collinearity genes between rice and sweet cherries are the least, indicating that the evolutionary relationship between rice as a monocot and sweet cherries was relatively distant. Some *PavARF* genes have more than one syntenic gene pair. For example, there are more than three pairs of collinearity genes between sweet cherry and soybean, and such as *PavARF13* up to five pairs (between apples and sweet cherry). It is speculated that these genes are highly conserved in evolution, and more replication events have occurred. Throughout, rice and sweet cherry genomes, few genes lacking collinearity gene pairs, for instance, *PavARF1/2/3*, and these gene pairs were found in other six dicotyledonous plants to sweet cherry, indicating that these orthologous pairs may be emerged after the separation of monocotyledonous and dicotyledonous plants. Also, there are some collinear gene pairs in all seven selected species, such as *PavARF7*. These gene pairs may come from the same ancestor and may already exist before monocotyledon and dicotyledon differentiation.

### 2.6. Expression Analysis of PavARF Genes in Stage-Specific and Tissue/Organ-Specific

Different tissues of the gene expression profile can let us to understanding and predicting biological functions. To characterize the expression pattern of the different members of the *PavARF* family, analyzed qRT-PCR data from 17 different samples: leaf, flower, fruit, stem, etc, and represented the expression levels using one heatmap (Figure 5a). The findings revealed that the expression level of each gene was divergent in different tissues, which further indicated that the ARF gene played a pivotal role in sweet cherry growth and development. Most of the members are highly expressed in fruit development and stems. The expression of *PavARF7/8/9/14* showed an up-regulated pattern as in collected flowers from four different developmental stages, while *PavARF4/15* was significantly down-regulated. In old leaves, *PavARF1/2/3/9/10/13* is highly expressed, which may be related to the aging and shedding of leaves. These genes may be involved in regulating leaf growth and development. *PavARF8* has the highest expression during the coloring period of fruit and may be involved in the regulation of fruit pigments. During the four stages from fruitlet to mature fruit, *PavARF5* showed up-regulated expression and reached the highest level in mature fruit. The expression of *PavARF6/7/12* gradually decreased. The expression level of *PavARF10* was highest in the second stage (FR2) of the fruit and then began to decrease. Compared with one-year-old stems (AS), *PavARF7/8/9/10* was up-regulated in two years old stems (BS), and *PavARF3/4/6/14* was down-regulated. In leaves and flower buds, most *PavARF* members maintain a low expression level.

As an auxin-responsive transcription factor, ARF usually interacts extensively with other transcription factors or proteins to regulate the growth and development of plants. Based on the studied *Arabidopsis thaliana* homologous AtARF protein, the ARF interaction mode of sweet cherry is predicted in the current study (Figure 5b). The results showed that there are interactions between ARFs and extensive interactions with the growth hormone response inhibitor IAA, which is consistent with previous studies. In addition, *PavARF* interacts with auxin-related genes such as TIR1, NPH4, AXR3 and ETT, indicating that *PavARF* has multiple functions in the auxin signal transduction pathway [31,32,33].

The physiological abscising of plant organs was a normal phenomenon. In order to explore the physiological abscising of sweet cherry fruitlet, we analyzed the differences in the expression of PavARF in the physiological fruit shedding at the fruitlet stage and carpopodiums. Compared with the non-abscission carpopodium, the expression level of *PavARF2/3/6/8/9/10/12/13/16* in the abscising carpopodium increased significantly, and *PavARF8/10* showed a very high expression level (Figure 5c). It indicates that these genes may be more sensitive to auxin and may also play an important role in the formation of the abscission zone.

In order to understand the expression of *PavARF* in abscising fruitlets, this study collected two physiological drops (0.5–0.7 cm, 1.3–1.6 cm), and collected normal fruits that did not fall off at the same time as a comparison to explore the expression of these genes. In the first physiological abscising fruitlet (FAb 1), compared with normal fruitlet, the expression of *PavARF10/13* was up-regulated significantly in the abscising fruit, while the expression of 11 members decreased (Figure 6). In the second physiological fruit abscission (FAb 2), the expression of *PavARF6/8/9/10/13/14/16* up-regulated expression, while *PavARF4/7/12/15* showed down-regulation. Compared with normal fruits, in the two physiological fruit drops, the gene whose expression level was up-regulated was *PavARF10/13*, and the gene whose expression was both down-regulated was *PavARF4/7/11/12/15*.

### 2.7. Expression of Low-Temperature and Drought Stress

The upstream promoter sequences of some members of *PavARF* contain *cis*-sequences associated with abiotic stress. *PavARF2/5/8/12/13/16* is an example that contained a low-temperature response component, while *PavARF1/5/7/10/11/16* was related to the induction of drought. To verify the changes of these genes under the stress of cold and drought, an analysis performed regarding their expression levels in these conditions. Under low temperature (Figure 7a), the expression level of *PavARF2/5/8/12/13/16* will increase within 1 h and was significantly higher than the control (25 ℃) without *PavARF12*, but it had a downward trend thereafter. The expression level inclined again at 24 h and then declined gradually. In addition, *PavARF8/12* has two low temperature response elements, and both genes have higher expression levels under low temperature. These two genes may be similar or complementary in function (overlapping functions), and they play an important regulatory role in the corresponding low temperature treatment [34].

For drought stress, 6 *PavARF* genes were selected for verification, and their upstream promoters have drought-responsive elements. The study found that under the drought treatment simulated by PEG, all 6 genes all demonstrated different degrees of response (Figure 7b). The expression level of *PavARF1* at 4 h was significantly higher compared to the control. *PavARF5/7/11* was down-regulated at 4–6 h after treatment, and the expression level was significantly lower than that of the control at the same time duration. Compared with the control, the expression level of *PavARF16* had no obvious change within 0–6 h after treatment, but it was significantly up-regulated at 8 h. The expression level of *PavARF10* was down-regulated at 2 h, and then tended to be up-regulated. There are two drought-responsive elements upstream of *PavARF7/10*, which may enhance their promoter activity.

## 3. Discussion

Auxins are vital regulators in plants. As a kind of transcription factor, ARFs participate in the signal pathways related to auxin response and regulate the growth and development of plants [35]. Our research has identified 16 ARF members in the sweet cherry genome. They are unevenly distributed on eight chromosomes and have highly similar domains. Compared with *Arabidopsis* 135 Mbp [36], rice 500 Mbp [37], tomato 828 Mbp [38], soybean 1115 Mbp [39], sweet orange 367 Mbp [40], apple 193 Mbp [41] and peach 265 Mbp [42], 260 Mbp sweet cherry genome is similar to Rosaceae peach (*Prunus persica*) [25]. The *PavARF* number is lower than 23 for *Arabidopsis*, 25 in rice [19], 21 for tomato [43], 51 in soybean [44], 19 in sweet orange [45] and 31 apples [30], but close to 17 for peach [17]. Through statistical analysis of the amino acids in the middle region of *PavARF*, the current study predicted that four *PavARF* genes have potential transcriptional activation activities and seven have potential transcriptional repressive activities. In *Arabidopsis*, the gene with transcriptional activation activity is *AtARF5/6/7/8/19*, and the others are transcription repressors [5]. Through domain analysis, we have found that most *PavARF* had B3 and auxin-resp domains, but it was found that *PavARF3* lacked the B3 domain. Such genes also have been founded in other species, such as *StARF18* in *Solanum tuberosum* [46]. The lack of the B3 domain meant that this gene could not recognize and bind to the auxin response element on the promoter of the target gene [16], but its CTD domain may combined with AUX/IAA or ARF [47]. As a transcription factor, ARF generally plays a role in the nucleus. Consistent with the subcellular location of ARF in other species, all *PavARF* members are predicted to be in the nucleus, proving that they function as transcription factors [17,48].

The ARF gene family is widely presented in land plants. From lower plants to higher plants, the ARF members have a clear tendency to enlarge and have relatively conservative homology relationships [49]. There is only one ARF in the bryophyte liverwort [50]. Compared with seven species such as *Arabidopsis thaliana*, rice has fewer collinearity gene pairs, indicating that ARF has undergone more extensive evolution and replication events after the staging of mono and dicot plants. The core signaling regions related to auxin response is conserved in angiosperms and bryophytes, and the molecular mechanism in these land plants is the same [51]. ARF in plants has been divided into three groups. Based on their transcriptional activity, groups A and B are defined as transcriptional activators and transcriptional repressors, respectively, and group C is recognized as transcriptional repressors based on the specific amino acids in the middle region (MR) [52]. According to evolutionary analysis, there are two ARF precursors in Charophytes, which are similar to A/B and C ARF, respectively. These results prove that the ARF gene may be derived from several common ancestor genes and begin to differentiate in algae plants [53].

The transcript expression analysis can help us understand the potentially distinct functions of *PavARF*. ARFs exhibit tissue-specific expression. *PavARF3/8/10* had a higher expression level in all tissues in sweet cherry. In rice, the 24 OsARFs did not have much difference in transcription level, indicating that ARF is constitutively expressed in rice [19]. During the germination of maize seeds, the expression level of more than half of *ZmARF* reached its peak after 24 h following imbibition and *ZmARF1* was the highest expressed in all tissues. The expression levels of 8 *ZmARF* in dry mature embryos were higher than immature embryos during seed development [18]. In sweet cherries, *PavARF8/10* is highly expressed in stems, while *PavARF15* is slightly expressed. Similarly, in soybeans, *GmARF12* had the highest expression in stems, while *GmARF19* has the lowest expression [44]. In *Tartary buckwheat*, the expression levels of 4 *FtARF* (*FtARF3/4/8/10*) in flowers were higher than in other groups, except for *FtARF7/18*, the expression levels of other genes in reproductive organs all had a higher level [54]. In the flower development of sweet cherry, *PavARF3/4/7/8/10/14* showed greater expression differences. *AtARF8* interacted with bHLH (basic helix-loop-helix) to influence petal growth by changed cell expansion or increased cell number [55]. *ARF6* and *ARF8* can promote inflorescence elongation and gynoecium development in tomato, as well as targeted by *miR167a*, which can down-regulate *ARF6* and *ARF8*. Furthermore, overexpression of *miR167a* in *Arabidopsis thaliana* can cause female infertility [56]. In peach fruit development, the expression of *PpARF12* tends to be stable, and the expression level in mature fruits decreases and *PpARF10A* is gradually up-regulated during the fruit development period, and the expression level reaches the maximum during maturity [17]. In sweet cherries, *PavARF8* was a more stable expression. This study indicated that ARF is widely involved in plant growth and development. 

Organ abscission is a part of the dynamic nature of the plant that involves changes in gene and cell function. In this study, *PavARF2/3/8/10/13* is up-regulated in the sweet cherry abscission fruitlets, and *PavARF6/7/11/12* is down-regulated (Figure 8). The study of ARF in plant organ abscising has been reported in other species. *AtARF2* and *AtARF* can promote the shedding of flowers of *Arabidopsis thaliana*, and have a higher expression in the base of the flower and the abscission zone and have pleiotropic effects on plant development [57]. In tomato, *miR160* was the main target gene of *SlARF10A*, and *miR160* mutation caused abnormal shedding of floral organs, accompanied by increased expression of *SlARF10A* [58]. Five ARFs expressions were down-regulated at 2 days or 4 days after the GPD (girdling plus defoliation) treatment in litchi fruitlet [59]. *LcARF2D/2E*, *7A/7B*, *9A/9B* and *16A/16B* likely acted as repressors in litchi fruit abscission, particularly *LcARF5A/B*, which might be positively involved in this process [24]. The expression of 22 ARF in *Citrus* did not show much difference in the process of abscission, indicating that ARF may not be involved in *Citrus* abscising fruits [60]. These studies indicate that there are great differences in ARF among different species.

## 4. Materials and Methods

### 4.1. Identification of ARF Genes in Cherry

Firstly, the sweet cherry genomic and protein data (*Prunus avium* Whole Genome Assembly v1.0 & Annotation v1 (v1.0.a1) were downloaded from the Rosaceae genome database (GDR, https://www.rosaceae.org/ (accessed on 22 February 2020)). Hidden Markov Model (HMM) files of ARF protein in Pfam (http://pfam.xfam.org/ (accessed on 22 February 2020)) were then obtained. The HMM files were used as queries to search against cherry protein sequences based on hmmsearch which is hmmer-3.2.1 version at the score value of 1 × 10^−20^ [61]. These sequences were used to construct the HMM model file of sweet cherry once the candidate sequences were acquired, and afterward, the model was used to search anew for all its (*Prunus avium*) proteins sequence. Upon discarded sequences that did not reflect the e-value, also could check whether the domain in the Conserved Domain Database (CDD, https://www.ncbi.nlm.nih.gov/Structure/cdd/wrpsb.cgi (accessed on 23 February 2020)) and Pfam databases were complete or correct and obtain all family members after eliminating the incomplete genes. These predicted members are named according to their position on the chromosome. The ExPASY online website (https://web.expasy.org/protparam/ (accessed on 23 February 2020)) was used to calculate the isoelectric point (pI), molecular weight (MW), Grand average of hydropathicity (GRAVY) and amino acid length of the ARF protein [62], and subcellular localization were forecasted by the online software Cell-PLoc-2.0 (http://www.csbio.sjtu.edu.cn/bioinf/Cell-PLoc-2/ (accessed on 24 February 2020)) [63].

### 4.2. Multiple Sequence Alignment and Phylogenetic Analyses

Clustal Omega (https://www.ebi.ac.uk/Tools/msa/clustalo/ (accessed on 25 February 2020)) was used for multiple sequence alignment of PavARF protein sequence. The phylogenetic trees were constructed using the ARF protein sequences of *Arabidopsis thaliana*, rice (*Oryza sativa*) and sweet cherry through MEGA7.0. Next, these trees were inferred using the Neighbor-Joining (NJ) method with the following 1000 bootstrap replications [64]. *Arabidopsis thaliana* and rice protein sequences were downloaded from the UniProt database (https://www.uniprot.org/ (accessed on 25 February 2020)). 

### 4.3. Sequence and Cis-Acting Element Analysis 

TBtools was applied to draw the exon-intron organization and conserved motifs of each *PavARF* gene by comparing the DNA and cDNA sequences corresponding [65]. To identify conserved motifs in PavARF proteins, the MEME (version 5.1.1, http://meme-suite.org/tools/meme (accessed on 2 March 2020)) online tool was tested. DNAMAN was applied to compute the amino acids in the PavARF MR region. According to the type of amino acid enrichment, whether it is a transcriptional activator or a transcriptional inhibitor was predicted. The upstream 2000 bp sequence of the *PavARF* gene was obtained from the *Prunus avium* genome and submitted to the PlantCARE website (http://bioinformatics.psb.ugent.be/webtools/plantcare/html/ (accessed on 7 March 2020)) to predict *cis*-acting elements [66]. From the prediction results, *cis*-acting elements related to response phytohormone and low temperature were selected for statistics.

### 4.4. Chromosomal Distribution and Gene Duplication of ARF Genes

Based on the genomic coordinates of the *Prunus avium* ARF genes retrieved from the GFF file, the MG2C (http://mg2c.iask.in/mg2c_v2.1/ (accessed on 10 March 2021)) was used to map genes on the chromosomes. MCScanX (Multiple Collinearity Scan toolkit) was adopted to analyze the gene duplication events, with the 1 × 10^−10^ parameters [67]. To exhibit the collinearity relationship of the ARFs obtained from sweet cherry and other selected species, the syntenic analysis maps were constructed using the TBtools.

### 4.5. Plant Materials

Seven years old “Brooks” sweet cherry trees planted under rain shelter coverings were used in this research. All materials were grown at the experimental orchard in the Key Laboratory of Plant Resource Conservation and Germplasm Innovation in Mountainous Region (Ministry of Education) of GuiZhou University in GuiYang WuDang Area (107°00′ E, 26°82′ N). Seventeen different tissue samples were used for tissue-specific expression pattern analysis. In detail, young leave (15 d leaf age), mature leave (45 d leaf age), old leave (170 d leaf age), one year old stem, two years old stem, 0.5–0.7 cm abscising fruit carpopodium and 1.3–1.6 cm abscising fruit carpopodium of the seven years old tree.

The flower samples were selected from bud dormancy (FL1), bud (FL2), before flowering (FL3) and blooming flowers with opened petals (FL4) of the seven years old trees. Fruit samples at different developmental stages were harvested at 11 (0.5–0.7 cm), 20 (1.3–1.6 cm), 32 (fruit colorings) and 44 (ripe fruits) days after anthesis of the seven years old trees, the 11, 20, 32 and 48 d were defined as FR 1–4, respectively. The abscission fruit samples involved 0.5–0.7 cm abscission fruits and 1.3–1.6 cm abscission fruits. Collected least 10 for each period fruit sample, and remove the fruit kernel, frozen in liquid nitrogen and stored at −80 °C for further experiments. 

For cold stress treatments, the sweet cherry twigs were subjected to 4 and 25 °C (control), respectively. The leaves were collected at 0, 1, 3, 6 and 12 h, and 1, 2 and 3 d in cold treatment. While as drought treatments, the sweet cherry twigs were soaked in 20% PEG6000 and dH_2_O (control). The leaf tissues from twigs were collected at 0, 2, 4, 6 and 8 h after treatment. All treated leaf samples were immediately frozen in liquid nitrogen and stored at −80 °C.

Total RNA from different organs was extracted using a polysaccharide polyphenol plant total RNA Extraction Kit (SENO, www.seno-bio.com, Zhangjiakou, Hebei Province, China). RNA was used for the synthesis of first strand of cDNA via PrimeScript^TM^ RT reagent Kit with gDNA Eraser (TaKaRa, Beijing, China). The qRT-PCR was carried out with the CFX Connect^TM^ Real-Time System instrument (BIO-RAD, Hercules, CA, USA) using PowerUp^TM^ SYBR^TM^ Green Master Mix (Thermo Fisher Scientific, Waltham, MA, USA). The sweet cherry *PavEF1-α2* and *PavRSP3* genes were used as an internal control. The sense and anti-sense primers were designed by Primer 5 and their specificity was detected by NCBI. Sequences of the primers used in this study were shown in detail in the Appendix A. The reactions were performed with three biological and technical replicates per sample. The data were analyzed using the 2^−ΔCt^ method [68]. STRING (https://string-db.org/ (accessed on 10 July 2020)) was used to analyze the interaction of PavARF proteins on the basis of the orthologs in *Arabidopsis*.

## 5. Conclusions

In the current study, 16 ARF members were identified from the *Prunus avium* genome, which were unevenly distributed on eight chromosomes. These gene members can be divided into five clades. Their gene structure is similar, and most members have a conserved ARF domain. These members are highly expressed during fruit development and may play an important role in fruit development. Among them, *PavARF10/13* is up-regulated expression in fruit drop, and *PavARF4/7/11/12/15* is down-regulated. These genes may be related to fruitlet abscising. Members with low temperature or drought response elements on the promoter can respond to low temperature or drought treatment.

## Figures and Tables

**Figure 1 ijms-22-11968-f001:**
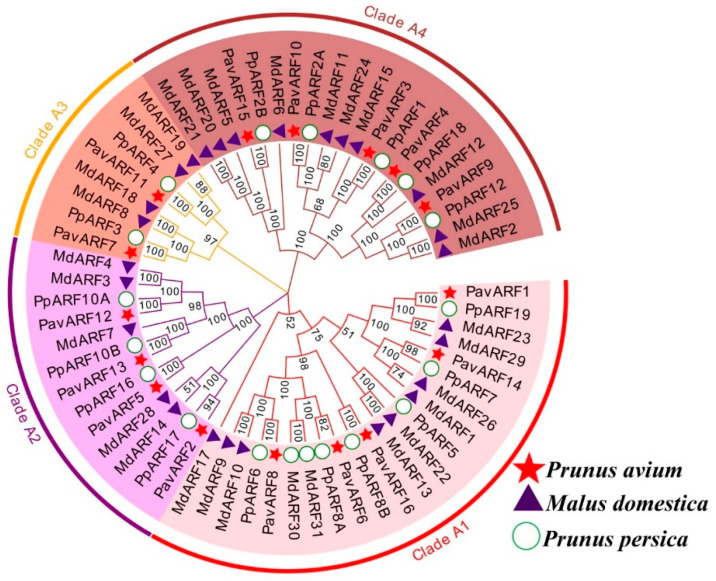
Phylogenetic tree of ARF proteins from sweet cherry, *Malus domestica*, and *Prunus persica*. All ARFs are divided into five clades, each signed by a different color, The PavARFs of the sweet cherry was marked red star.

**Figure 2 ijms-22-11968-f002:**
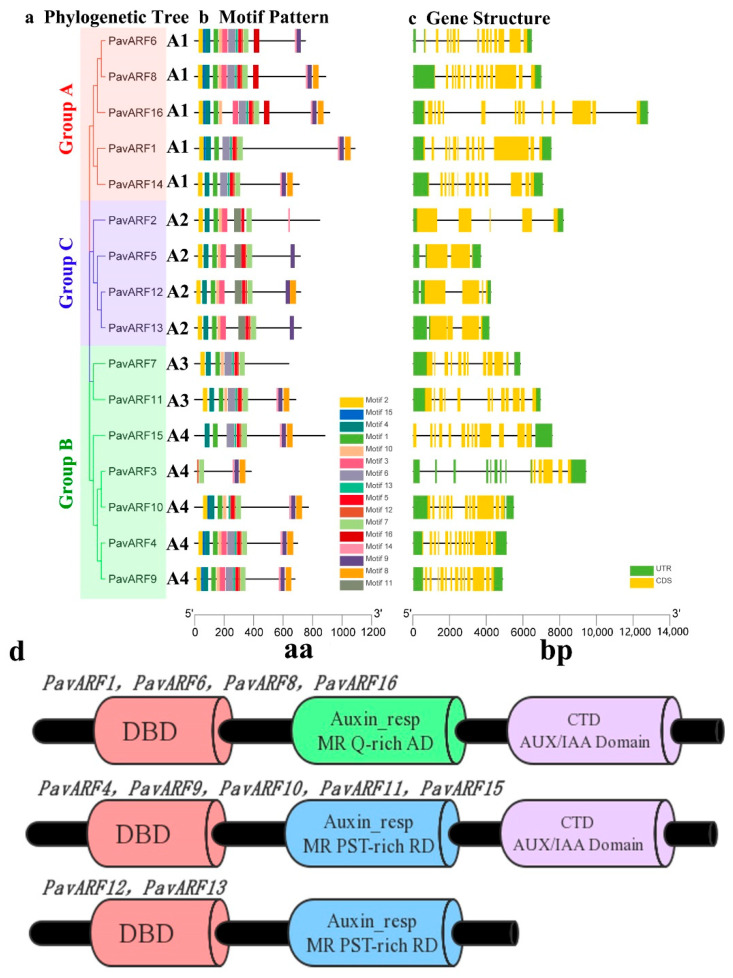
The structure distribution patterns of 16 *PavARF* genes. (**a**). PavARF phylogenetic tree, clustered into three Group (Group A, B and C). (**b**). The discovered conserved motifs of PavARF proteins. 16 Motifs were found and illustrated by a different color. (**c**). Exon-intron structures of *PavARF* genes. Exons and introns are shown as yellow color and black lines. The left and right end ware untranslated regions and illustrated by green color. (**d**). The transcriptional activator or transcriptional inhibitor of *PavARF* predicted by DNAMAN.

**Figure 3 ijms-22-11968-f003:**
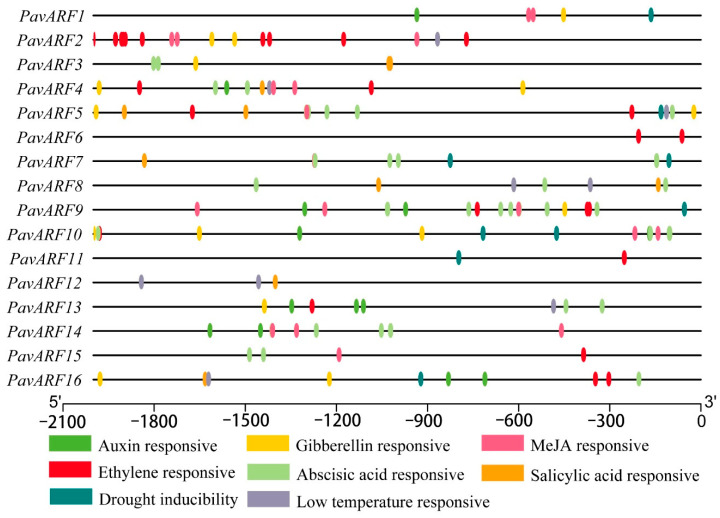
Analysis of specific *cis*-elements in *PavARF* promoters. The 2000 bp upstream sequences were used to analyze six specific phytohormone-related cis-elements (auxin, gibberellin, MeJA, ethylene, abscisic acid and salicylic acid), two stress-responsive regulatory elements including drought and low-temperature. They are illustrated by the different color boxes.

**Figure 4 ijms-22-11968-f004:**
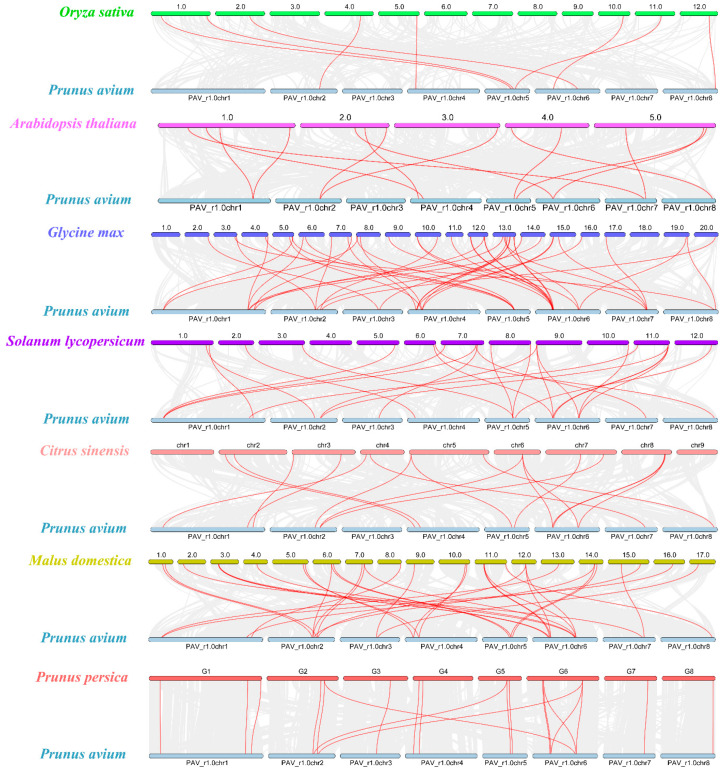
Collinearity analyses of ARF genes between *Oryza sativa*, *Arabidopsis thaliana*, *Glycine max*, *Solanum lycopersicum*, *Citrus sinensis*, *Malus domestica* and *Prunus persica*. Gray lines in the background indicate the collinear blocks within the Prunus avium and other plant genomes, while the red lines highlight the syntenic ARF gene pairs. Different species are marked with different colors, and sweet cherry is marked with light blue.

**Figure 5 ijms-22-11968-f005:**
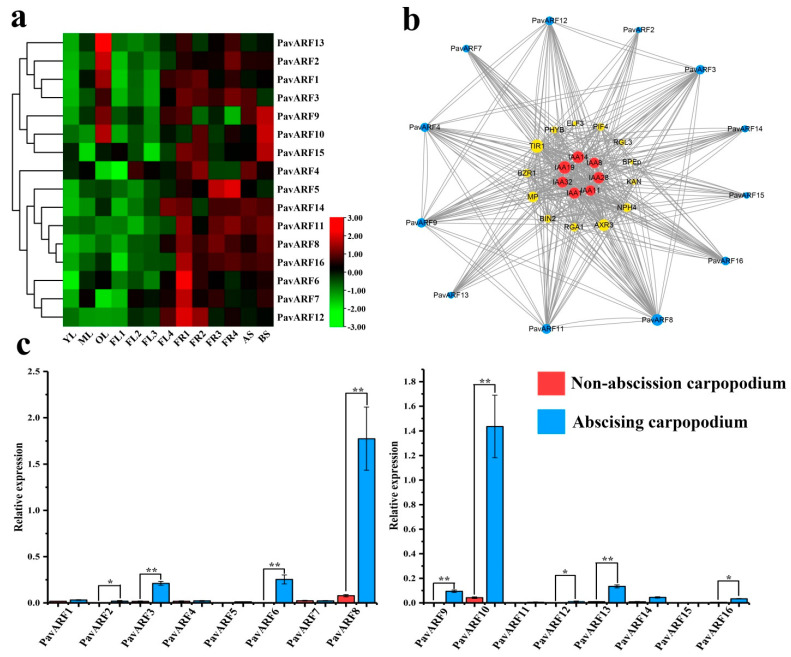
(**a**) Relative mRNA expression levels of *PavARF* genes in the different sweet cherry tissue. Note: Young Leaves (YL), Mature Leaves (ML), Old Leaves (OL), Dormant Flower bud (FL1), Flower bud (FL2), Blooming Flowers (FL3), Florescence (FL4), Small Fruit (FR1), Middle Fruit (FR2), Fruit of Red-fleshed (FR3), Ripe Fruit (FR4), Annual Stem (AS), Biennial Stem (BS). (**b**) Predicted protein-protein interaction networks of PavARF proteins using STRING server. (**c**) Expression pattern of *PavARF* genes in sweet cherry carpopodium. * represents *p* < 0.05 in the variance analysis and ** represents *p* < 0.01 in the variance analysis.

**Figure 6 ijms-22-11968-f006:**
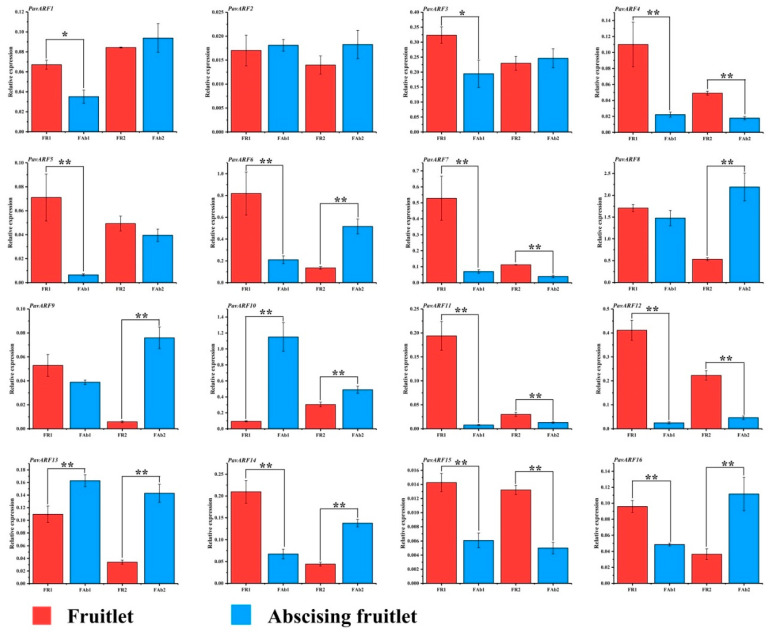
Expression levels of PavARFs at the sweet cherry fruitlet and abscising fruitlet. * represents *p* < 0.05 in the variance analysis and ** represents *p* < 0.01 in the variance analysis.

**Figure 7 ijms-22-11968-f007:**
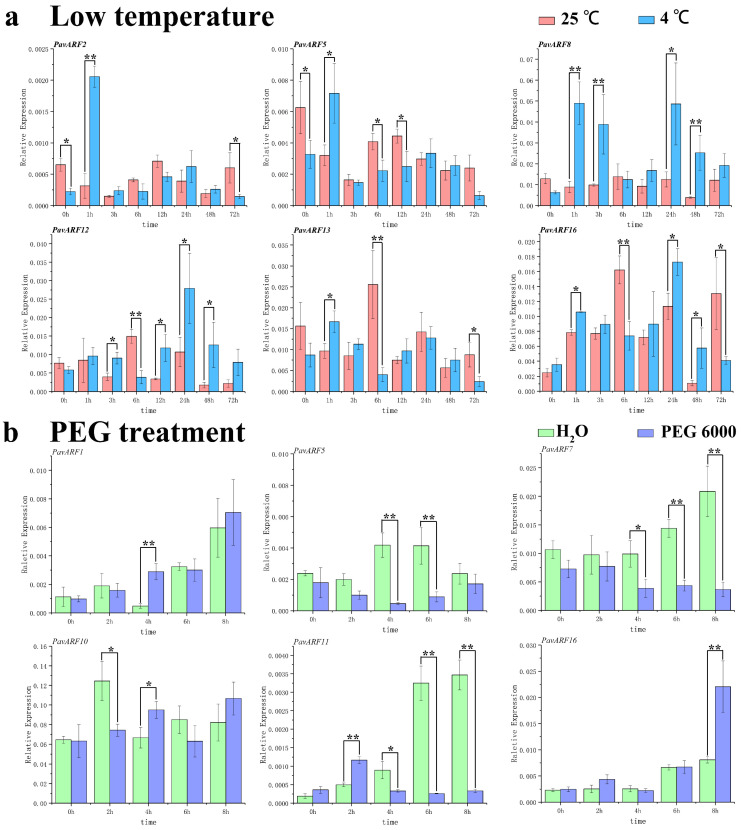
Expression profiling of *PavARF* at the low-temperature stress (**a**) and drought (**b**). * represents *p* < 0.05 in the variance analysis and ** represents *p* < 0.01 in the variance analysis.

**Figure 8 ijms-22-11968-f008:**
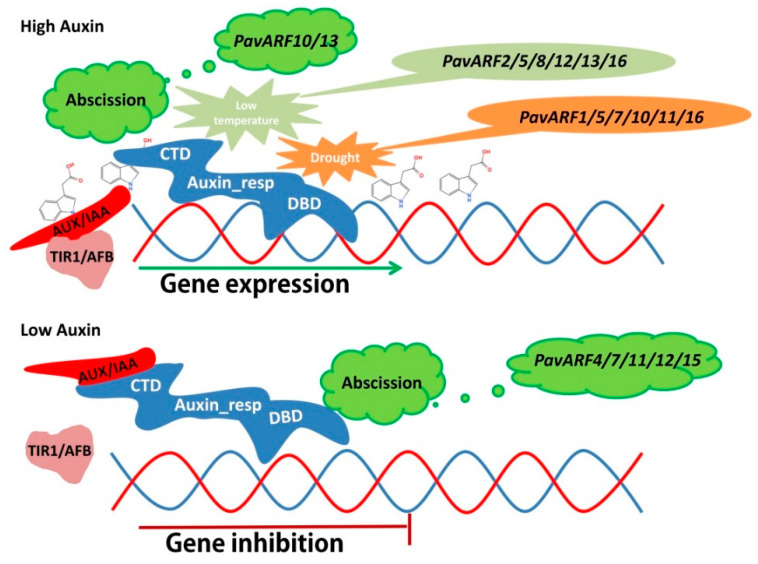
The expression patterns of sweet cherry ARF gene family members during fruit shedding, drought and low temperature stress. Under drought, low temperature and shedding, PavARF binds to AuxRE elements under the action of auxin to initiate gene expression. For other genes, when the auxin is low, PavARF binds to AUX/IAA, and the gene expression is inhibited.

**Table 1 ijms-22-11968-t001:** Basic information of the ARFs gene family in sweet cherry.

Gene Name	Gene ID	CDS	Protein	*M*w (KDa)	pI	GRAVY	Subcellular Localization	Domain
*PavARF1*	Pav_sc0001248.1_g270.1.mk	3264	1187	119627.80	5.86	−0.608	Nucleus	B3,Auxin_resp,CTD
*PavARF2*	Pav_sc0000030.1_g880.1.mk	2547	848	93909.05	6.96	−0.449	Nucleus	B3,Auxin_resp,CTD
*PavARF3*	Pav_sc0001196.1_g1840.1.mk	1155	384	42848.91	5.20	−0.662	Nucleus	Auxin_resp,CTD
*PavARF4*	Pav_sc0000586.1_g230.1.mk	2097	698	77581.78	6.44	−0.430	Nucleus	B3,Auxin_resp,CTD
*PavARF5*	Pav_sc0000084.1_g430.1.mk	2151	716	79165.21	6.59	−0.408	Nucleus	B3,Auxin_resp
*PavARF6*	Pav_sc0001345.1_g010.1.mk	2256	751	82327.61	5.90	−0.335	Nucleus	B3,Auxin_resp,CTD
*PavARF7*	Pav_sc0001900.1_g150.1.mk	1920	639	69349.71	5.92	−0.530	Nucleus	B3,Auxin_resp
*PavARF8*	Pav_sc0003135.1_g100.1.mk	2670	889	98481.69	6.08	−0.431	Nucleus	B3,Auxin_resp,CTD
*PavARF9*	Pav_sc0002250.1_g170.1.mk	2046	681	75315.71	6.20	−0.476	Nucleus	B3,Auxin_resp,CTD
*PavARF10*	Pav_sc0000042.1_g450.1.mk	2316	771	86014.22	5.85	−0.694	Nucleus	B3,Auxin_resp,CTD
*PavARF11*	Pav_sc0000094.1_g590.1.mk	2061	686	75602.80	5.35	−0.499	Nucleus	B3,Auxin_resp,CTD
*PavARF12*	Pav_sc0002446.1_g140.1.mk	2160	719	79093.28	7.88	−0.410	Nucleus	B3,Auxin_resp
*PavARF13*	Pav_sc0000711.1_g070.1.mk	2172	723	79392.80	6.09	−0.413	Nucleus	B3,Auxin_resp
*PavARF14*	Pav_sc0000129.1_g1480.1.mk	2130	709	78539.30	5.34	−0.444	Nucleus	B3,Auxin_resp,CTD
*PavARF15*	Pav_sc0000848.1_g300.1.mk	2652	883	97861.21	5.92	−0.444	Nucleus	B3,Auxin_resp,CTD
*PavARF16*	Pav_sc0001314.1_g050.1.mk	2751	916	101887.61	5.76	−0.410	Nucleus	B3,Auxin_resp,CTD

## Data Availability

Not applicable.

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
