# Peer review of "Genome-Wide Identification of ARF Gene Family Suggests a Functional Expression Pattern during Fruitlet Abscission in Prunus avium L."

_ijms, 2021, doi:10.3390/ijms222111968_

Round 1

Reviewer 1 Report

Dear Authors, I have included all my comments and suggestions directly in the *.pdf attached.

Reviewer 2 Report

In this manuscript authors did the genome-wide identification of the ARF gene family suggests a functional expression pattern during fruitlet abscission in Prunus avium L. In the current study, 16 ARF members have been identified in the sweet cherry genome. These genes are all located in the nucleus. Sequence analysis showed that genes in the same subgroup have similar exon-intron structures. A phylogenetic tree has been divided into 5 groups. The promoter sequence includes 6 kinds of plant hormone-related elements, as well as abiotic stress response elements such as low temperature or drought. The expression patterns of PavARF in different tissues, fruitlet abscission, cold and drought treatment were comprehensively analyzed. PavARF10/13 was up-regulated and PavARF4/7/11/12/15 was down-regulated in fruitlet abscising. These genes may be involved in the regulation of fruit drop in sweet cherry fruits.

The manuscript is written very well but for the betterment of this manuscript I have few suggestions:

  1. The introduction is short. The author should include recent genome-wide studies such as: a.Genome-Wide Identification and Characterization of PIN-FORMED(PIN) Gene Family Reveals Role in Developmental and Various Stress Conditions in Triticum aestivum. b. Genome-wide identification and expression pattern analysis of the KCS gene family in barley. C. Genome-wide identification and characterization of abiotic stress responsive lncRNAs in Capsicum annuum. d. Genome-Wide Identification and Characterization of the Brassinazole-resistant (BZR) Gene Family and Its Expression in the Various Developmental Stage and Stress Conditions in Wheat (Triticum aestivum). e. Genome-wide identification and functional characterization of natural antisense transcripts in Salvia miltiorrhiza. f. Genome-wide identification and expression analysis of the AT-hook Motif Nuclear Localized gene family in soybean.

2. It would be better if the author tries to validate the function of at least one gene they found in this study.

3. Make one hypothetical figure which depicts the findings of this study.

Change at

L215 more than one syntenic gene pairs to more than one syntenic gene pair.

L300 response and regulated the growth to response and regulates the growth.

L315 this gene cannot recognize to this gene could not recognize.

L382 Five ARFs expression to Five ARFs expressions.

L388 were greatly differences to were great differences.

L400 Pfam databases is complete to Pfam databases are complete.

L401 members after eliminated the incomplete genes to members after eliminating the incomplete genes.

I have detected plagiarism in this manuscript at L85-87, L96, L140-143, L148-149, L158-159, L209, L221, L319-320, L342-343, L371-373, L451-452,

Round 2

Reviewer 2 Report

The manuscript has improved a lot. It can be accepted after minor changes.

  1. The author did not add all the references I have suggested. Please add all of them.
  2. The last figure should be in discussion not in the conclusion part. 

Author Response

Thank you for reviewing our manuscript again. Your comments and suggestions are very valuable and helpful for revising and improving our manuscript. The following is our response to your comments point-by-point.

  1. The author did not add all the references I have suggested. Please add all of them.

Response: We are very grateful for your suggestions. These articles you recommend are of great reference value, and we have cited one of them in the newly submitted manuscript. The ideas and experimental design of these articles are very complete, and we will consider citing them in the next work. Thank you for reviewing our manuscript and suggestions, sincerely hope you can understand. In view of the field covered by this research, we will not cite other articles. Thank you again for reviewing this article.

  1. The last figure should be in discussion not in the conclusion part.

Response: Thank you for your suggestion. We have adjusted the position of Figure 8 in the newly uploaded manuscript. Put Figure 8 in the discussion section.